Journal of Machine Learning Research 2 (2025) 1-11        Submitted 6/30; Revised; Published

# Reliable and Efficient Tissue Segmentation in Whole-Slide Images

**Sander Elias Magnussen Helgesen**[1]      SANDER@ICGI.NO
[1] *Institute for Cancer Genetics and Informatics, Oslo University Hospital, Oslo, Norway*
**Anthony Manet**[1]      ANTHONY@ICGI.NO
**Karolina Cyll**[1]      KAROLINA@ICGI.NO
**Kari Anne Risan Tobin**[1]      KARIANNE@ICGI.NO
**Marna Lill Kjæreng**[1]      MARNA@ICGI.NO
**Ilyá Kostolomov**[1]      TECH@ICGI.NO
**Audun Ljone Henriksen**[1]      AUDUN@ICGI.NO
**Sepp de Raedt**[1]      SEPP@ICGI.NO
**Hanne Arenberg Askautrud**[1]      HANNE@ICGI.NO
**Miangela Lacle**[2]      M.M.LACLE@UMCUTRECHT.NL
[2] *University Medical Center Utrecht, Utrecht, Netherlands*
**Robert Jones**[3]      ROBJONES@LIVERPOOL.AC.UK
[3] *Department of Molecular and Clinical Cancer Medicine, University of Liverpool, Liverpool, United Kingdom*
**Cornelis Verhoef**[4]      C.VERHOEF@ERASMUSMC.NL
[4] *Erasmus MC Cancer Institute, University Hospital Rotterdam, The Netherlands*
**Tarjei Sveinsgjerd Hveem**[1]      TARJEI@ICGI.NO
**Ole-Johan Skrede**[1]      OLE-JOHAN@ICGI.NO
**Andreas Kleppe**[1,5,6]      ANDREKLE@IFI.UIO.NO
[5] *Department for Informatics, University of Oslo, Oslo, Norway*
[6] *Centre for Research-based Innovation Visual Intelligence, UiT The Arctic University of Norway, Tromsø, Norway*

**Editor:**

## Abstract

Whole-slide images in digital pathology often contain large regions of irrelevant background, making tissue segmentation an important preprocessing step in many applications. Traditional rule-based approaches to tissue segmentation often work quite well, but it is difficult to create general rules that cover all instances. We here apply an unmodified nnU-Net v2 training setup on downsampled whole-slide to develop and test an efficient and robust tissue segmentation model. The dataset contained nearly 30 000 images from slides with different tissue types, imaged using different scanners, and annotated using a semi-automatic workflow so that all annotations have been verified or made by human experts. This large, diverse dataset enables the training of a tissue segmentation model that generalizes well across different scanners and tissue types. We observed that our proposed model achieves similar or better accuracy than other deep learning models, while offering better robustness than simpler rule-based methods. The best compromise between inference speed and accuracy was observed using images at 10 µm per pixel. Our approach can be used as an efficient and well-suited preprocessing step for computational pathology. Source

code, Dockerfiles, and model weights are made publicly available at: `https://github.com/icgi/Reliable-and-Efficient-Tissue-Segmentation-in-Whole-Slide-Images`.

**Keywords:** Machine learning, Image segmentation, Pathology, nnU-Net, Deep learning

## 1 Introduction

Whole-slide images (WSIs) in digital pathology routinely exceed $100\,000{\times}50\,000$ pixels at full resolution, which can make tissue segmentation a computationally burdensome pre-processing step before further analysis. Classical rule-based methods, ranging from global thresholding to stain-specific color deconvolution, are fast, but struggle to generalize across different scanners, staining types, and tissue types, often requiring manual tuning for each new dataset.

In this work, we demonstrate that unmodified nnU-Net v2 (Isensee et al. (2021)), originally developed for radiology tasks, also works well on downsampled WSIs in pathology. To train the network and evaluate its performance, we curated a large, diverse dataset with over $28\,000$ images. We evaluate our pipeline against both rule-based baselines and state-of-the-art methods, showing that our streamlined approach matches or exceeds segmentation accuracy while substantially cutting end-to-end processing time. **Our main contributions are as follows:** (1) we demonstrate that the unmodified nnU-Net v2 achieves high-quality tissue segmentation, without sacrificing inference speed; (2) we publish a simplified, easy-to-use pipeline, and the trained model weights; (3) we analyze speed-accuracy trade-offs by repeating the experiment at different input resolutions.

## 2 Materials and methods

### 2.1 Dataset preparation

In total, we put together a dataset consisting of $28\,858$ WSI scans from seven diverse pathology projects, covering multiple scanner vendors, laboratories, tissue types, and nationalities (Table 1). Hematoxylin and Eosin (H&E) staining was used for all slides. All scans were shuffled and split 80/20 into a training set with $23\,086$ scans and a test set containing $5\,772$ scans. Our dataset uses 3 different scanners: Aperio AT2 (Leica, Germany), NanoZoomer XR (Hamamatsu, Japan), and 3D-Histech (Pannoramic, Hungary).

Due to limitations in hardware and the input requirements of the model architecture, we downsampled the raw WSIs to 10 µm per pixel. This significantly reduced file size from several gigabytes per scan to approximately 1–10 MB, while preserving sufficient detail for tissue segmentation. These downsampled images were used both for creating the annotations and training the neural networks.

### 2.2 Annotation workflow

To efficiently generate tissue-background annotations to use for model development and evaluation, we used a semi-automated workflow. First, we applied automatic tissue segmentation methods to produce the initial annotation masks. These initial masks were then manually reviewed to ensure quality and consistency. Out of $28\,858$ scans, $2\,584$ were identified as having inadequate annotations. These scans were re-evaluated by domain experts at

| Source institution | Tissue type | Scanner | Patients | Scans |
|---|---|---|---|---|
| University of Liverpool | CRLM | AT2, XR | 151 | 2 952 |
| Erasmus University Medical Center | CRLM | P1000 | 960 | 8 388 |
| National Cancer Center Hospital East | CRC | XR | 116 | 522 |
| University College London Hospitals | CRP | AT2, XR | 3 000 | 15 707 |
| University of Leeds | CRP | AT2 | 291 | 299 |
| University Medical Center Utrecht | CRP | AT2, XR | 308 | 990 |
| Total count | | | 4 826 | 28 858 |

Table 1: Summary of datasets used in this study. CRC = colorectal cancer; CRP = colorectal polyp; CRLM = colorectal liver metastasis; AT2 = Aperio AT2; XR = NanoZoomer XR; P1000 = Pannoramic 1000

the Institute for Cancer Genetics and Informatics at Oslo University Hospital, who manually corrected errors where the automated methods had failed to accurately mask the tissue. The corrected masks were then merged with the initially accepted ones to form the final annotation set used for training and evaluation.

## 2.3 Neural network architecture

To develop a robust and efficient pipeline for background tissue segmentation, we used the standard nnU-Net v2 framework (Isensee et al. (2021)) from the GitHub repository `https://github.com/MIC-DKFZ/nnUNet` (commit hash: ac79a61). This architecture is designed to work on various medical segmentation tasks. It automatically configures most training parameters, e.g., input patch size and batch size, based on dataset characteristics such as image dimensions, intensity distribution, and available computational resources.

In addition to the standard nnU-Net v2 architecture, we evaluated the residual encoder (ResEnc) variant of nnU-Net (Isensee et al. (2024)), which adds residual blocks to help the model learn better features and generalize better across harder segmentation tasks. This architecture has demonstrated improved segmentation accuracy on several datasets, though with the tradeoff of being more resource-demanding and with a slight increase in inference time.

## 2.4 Baseline comparisons

For benchmarking, we compared our models against both rule-based and deep learning-based segmentation methods. Rule-based baselines include Otsu's method and two in-house algorithms representing common simple approaches. For deep learning baselines, we evaluated against the tissue segmentation method implemented in the PathProfiler pipeline (Haghighat et al. (2022)) and test the performance of one of the suggested nnU-Net configurations proposed by Spronck et al. (2023).

### 2.4.1 Non-deep learning based segmentation

As a reference point, we employed three purely rule-based segmentation pipelines that rely on classical processing primitives. These methods build on classical computer vision

techniques such as morphological filtering, edge detection, and thresholding. They are fast and predictable, and serve as useful baselines for evaluating more advanced (e.g., machine learning-based) segmentation models.

**Otsu's method.** We applied Otsu's thresholding method (Otsu et al. (1975)) to the 10 µm per pixel images, preceded only by a 5×5 smoothing filter to reduce noise. No additional tuning was performed.

**Canny-based segmentation.** This method works on 5 µm per pixel images and uses Canny's algorithm (Canny (1986)) to detect edges in the input image. This is followed by morphological closing with a 9×9 structuring element and filling of holes smaller than 10 000 pixels, before morphological opening and removal of foreground regions smaller than 1 600 pixels.

**Intensity-based segmentation.** This method converts RGB images to grayscale HSV color space and segments tissue based on transforming intensities using percentiles and thresholding the values. This process is followed by morphological closing and opening in combination with removal of 4-connected background and foreground regions. A resolution of 16.3745 µm per pixel was used for the input images.

### 2.4.2 DEEP LEARNING BASED SEGMENTATION

We compare against two deep learning based segmentation methods: PathProfiler segmentation and Pathology nnU-Net.

**PathProfiler**. We used the tissue segmentation model from the PathProfiler pipeline, introduced by Haghighat et al. (2022), as a deep learning baseline. PathProfiler uses a U-Net-based architecture (Ronneberger et al. (2015)) for background tissue segmentation and is designed to operate directly on WSIs. To ensure compatibility with our test data, we implemented a data reader class to support the MRXS format used by 3DHISTECH scanners. No changes were made to the model weights or its core configuration. PathProfiler was initialized and evaluated using the pretrained model weights provided by Haghighat et al. (2022), without further tuning. For fair comparison, we provided WSIs as input instead of PNG images and let the PathProfiler pipeline handle downsampling to its desired resolution. All pipeline parameters were left unchanged.

**Pathology nnU-Net**. nnU-Net for pathology by Spronck et al. (2023) is an initiative to simplify and create a pipeline design for straight forward use of histopathology images with the nnU-Net pipeline. The authors introduce a dynamical data loader to load the WSIs on the fly, avoiding the memory issues related to the sheer size of WSIs, as well as other small improvements, making it straightforward to use for pathology. In addition, the authors suggest several modifications to the initial parameters that better fit pathology images.

While our approach shares goals with the Pathology nnU-Net framework, we applied a simplified pipeline that avoids modifying the original code pipeline. For comparison, we evaluated one of their reported high-performing configurations (Config E), which closely matches our setup and can be performed using the standard nnU-Net pipeline. Config E uses RGB to 0-1 normalization instead of Z-score normalization, a fixed batch size of 8, and a patch size of 512. We trained this configuration on the same data split as our main experiments, without any further tuning.

## 2.5 Evaluation metrics

We use the Dice score (Sørensen (1948)) and IoU (Intersection over Union) averaged across datasets to systematically evaluate model performance. Both the Dice score and IoU measure the overlap between the predicted mask and ground truth and are computed per scan. We also analyze the distribution of Dice scores across the dataset and report score thresholds in 10% increments to visualize the model's performance.

## 3 Results

We evaluated segmentation accuracy and inference time across multiple resolutions and model variants. Comparisons are made against classical rule-based methods and deep learning baselines. All models are evaluated on our test set of 5 772 image overviews of WSIs taken from various datasets.

All models were trained on an NVIDIA RTX 3090 GPU with 24 GB of memory. We use the official nnU-Net v2 implementation with default settings, and no architectural modifications or custom preprocessing steps were applied. Each network was trained for 1 000 epochs, with the standard nnU-Net averaging 14.9 GPU-hours and consuming 4.87 kWh. The ResEnc nnU-Net required 59.75 GPU-hours, with an overall energy consumption of 21.84 kWh. For fair benchmarking, input scans were resized to match the intended resolution and format of each method. Additionally, we do not perform any extra post-processing to maintain consistency with other methods. A Docker project containing all the requirements and versions used in our experiments is included in our GitHub repository.

### 3.1 Inference time

We quantified the trade-off between resolution, speed, and accuracy by running nnU-Net and its residual-encoder variant at 5, 10, and 20 µm per pixel, and compared against PathProfiler (Table 2). PathProfiler works at different resolutions, but for simplicity we use the default 8 µm per pixel resolution. Elapsed inference times were averaged over 100 random WSIs from different scanners and tissues, while the Dice and IoU scores were averaged over all 5 772 WSIs. ResEnc yielded modest but consistent accuracy gains over the original nnU-Net at increased computational cost. The two models trained at 5 µm per pixel performed similarly to the two models trained at 10 µm per pixel but required substantially more inference time. While the 20 µm model offered about a threefold speed gain, visual inspection revealed an increased number of segmentation outliers caused by artifacts and loss of details (see supplementary Figure S1). Therefore, the two models trained on 10 µm per pixel resolution were considered to be the best trade-off between inference speed and segmentation accuracy, and were used in the further evaluations described below. Our model runs at 10 µm versus PathProfiler's 8 µm resolution, so downsampling the WSIs before inference should be slightly faster for us.

### 3.2 Masking results

Figure 1 shows the distribution of Dice and IoU scores across all models. Our nnU-Net models outperformed both rule-based baselines and PathProfiler, and showed similar results as the Pathology nnU-Net variants. Visual comparisons of our nnU-Net models trained at

| Resolution (µm/px) | Model | Dice score (%) | IoU score (%) | Inference time (s) |
|---|---|---|---|---|
| 5 | nnU-Net | 98.52 | 97.25 | 5.88 |
| 5 | ResEnc | 98.97 | 98.06 | 11.70 |
| 10 | nnU-Net | 98.54 | 97.26 | 1.42 |
| 10 | ResEnc | 98.96 | 98.02 | 3.09 |
| 20 | nnU-Net | 98.54 | 97.23 | 0.44 |
| 20 | ResEnc | 98.81 | 97.71 | 0.89 |
| 8 | PathProfiler | 94.51 | 89.94 | 2.25 |

Table 2: Segmentation performance and runtime for different resolutions.

10 µm per pixel vs the PathProfiler model suggest that the latter is less sensitive to detect tissue with faint appearance, such as mucinous tissue (see Figure 2 for an example).

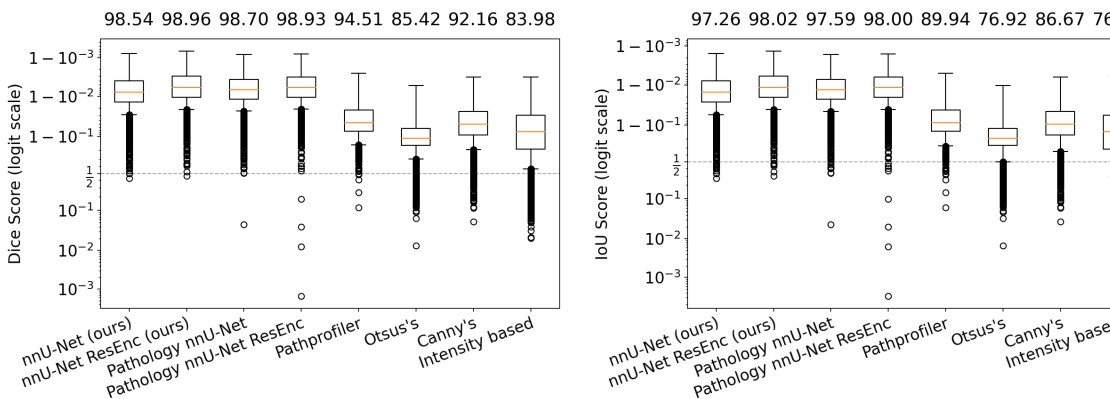

(a) Average Dice scores presented as percentages above plot.

(b) Average IoU scores presented as percentages above plot.

Figure 1: Distribution of Dice score (on the left) and IoU score (on the right) for different models.

To investigate how often each segmentation approach fails to properly segment the tissue, we evaluated the proportion of WSIs in the test set above specific Dice score thresholds. The ResEnc nnU-Net model achieved Dice scores above 90% on over 99% of the WSIs, while PathProfiler exceeded this threshold on just under 95% of the WSIs (Figure 3). This reflects the increased average Dice score of our nnU-Net models, which is mainly related to a coarse segmentation of tissue that match a manual annotation better. However, Path-Profiler additionally exhibited several outlier cases with Dice scores below 50%, indicating that PathProfiler fails to segment tissue more frequently than our nnU-Net models, which were never observed to give this low Dice score. Also, the Pathology nnU-Net models were observed to result in a few cases with very low Dice score. The rule-based baselines failed to segment tissue properly more often than the deep learning-based models. From supplementary Figure S2, we see a case where all rule-based baselines and PathProfiler, oversegment, and fail to separate the tissue from the gray regions with fecal matter.

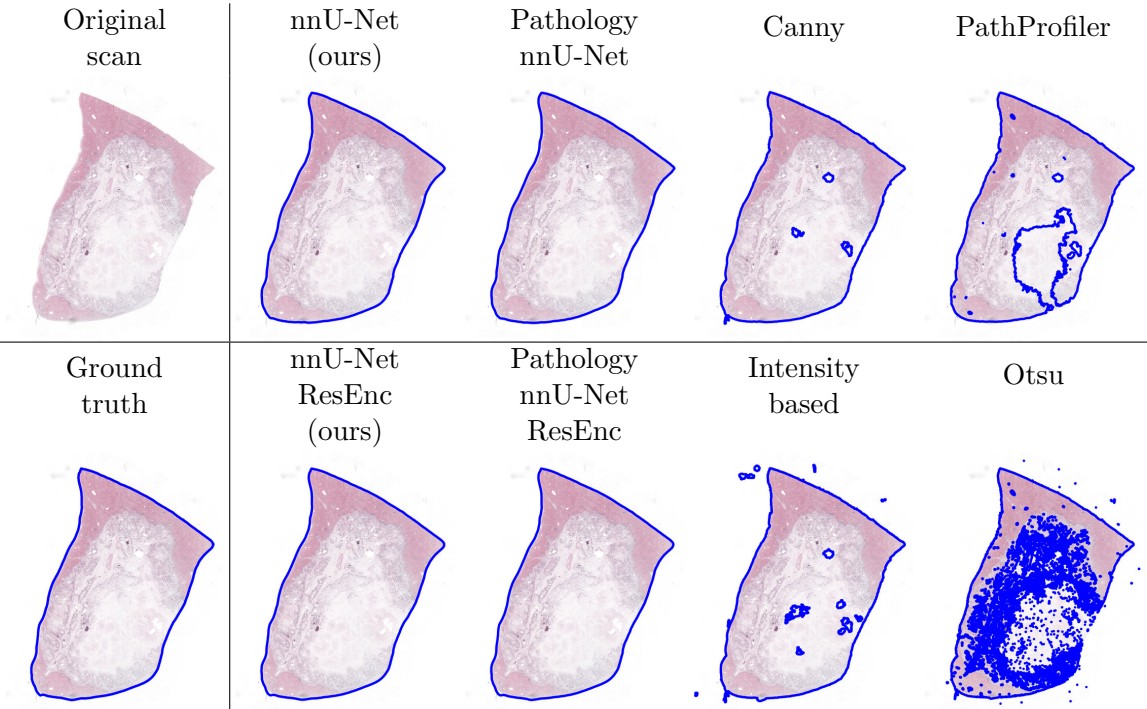

Figure 2: Blue outlines of segmentations generated by different approaches when applied to a WSI showing mucinous tissue.

## 4 Conclusion

We evaluated the tissue segmentation performance of an unmodified nnU-Net v2 pipeline in digital pathology. Using a varied dataset of over 28 000 WSIs, we observed that our nnU-Net model obtained an average Dice score of 98.97% and an average IoU score of 98.02%. In comparison, the rule-based segmentation methods performed worse on average than the deep learning variants, and also had substantially more failed segmentations with a Dice score below 50%. The U-Net from PathProfiler showed a lower Dice and IoU score than the nnU-Net variants, which, in turn, were quite similar. Furthermore, an analysis of inference time showed that we could obtain a good compromise between accuracy and inference speed with a resolution of 10 µm per pixel. The results are based exclusively on H&E-stained slides, and assessing the model's generalizability to other stains, such as IHC, remains an open direction for future work.

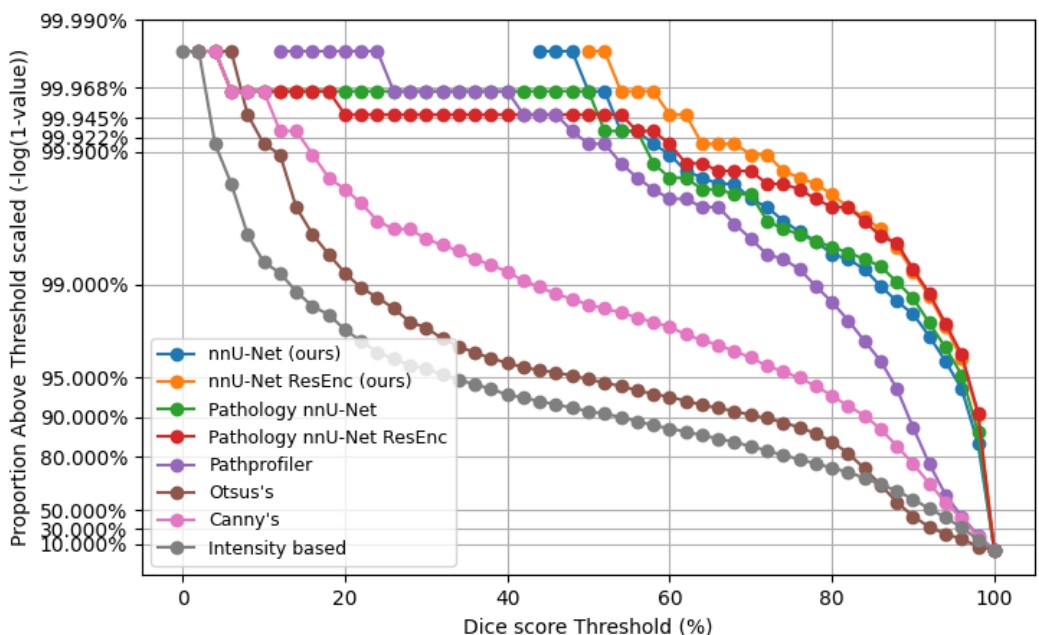

Figure 3: Proportion of WSIs with Dice score above specific thresholds for different segmentation approaches.

## Acknowledgments and Disclosure of Funding

We thank Krishanthi Harikaran, Ingrid Elise Weydahl, and Maria Isaksen for laboratory assistance. We are also grateful to Zhen Qian for facilitating the acquisition of the dataset from the Erasmus University Medical Center Cancer Institute. This work was supported by the South-Eastern Norway Regional Health and Authority research fund (grant number 2024039) and The Norwegian Cancer Society (grant number 273051).

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

## Appendix A. Tissue segmentation examples

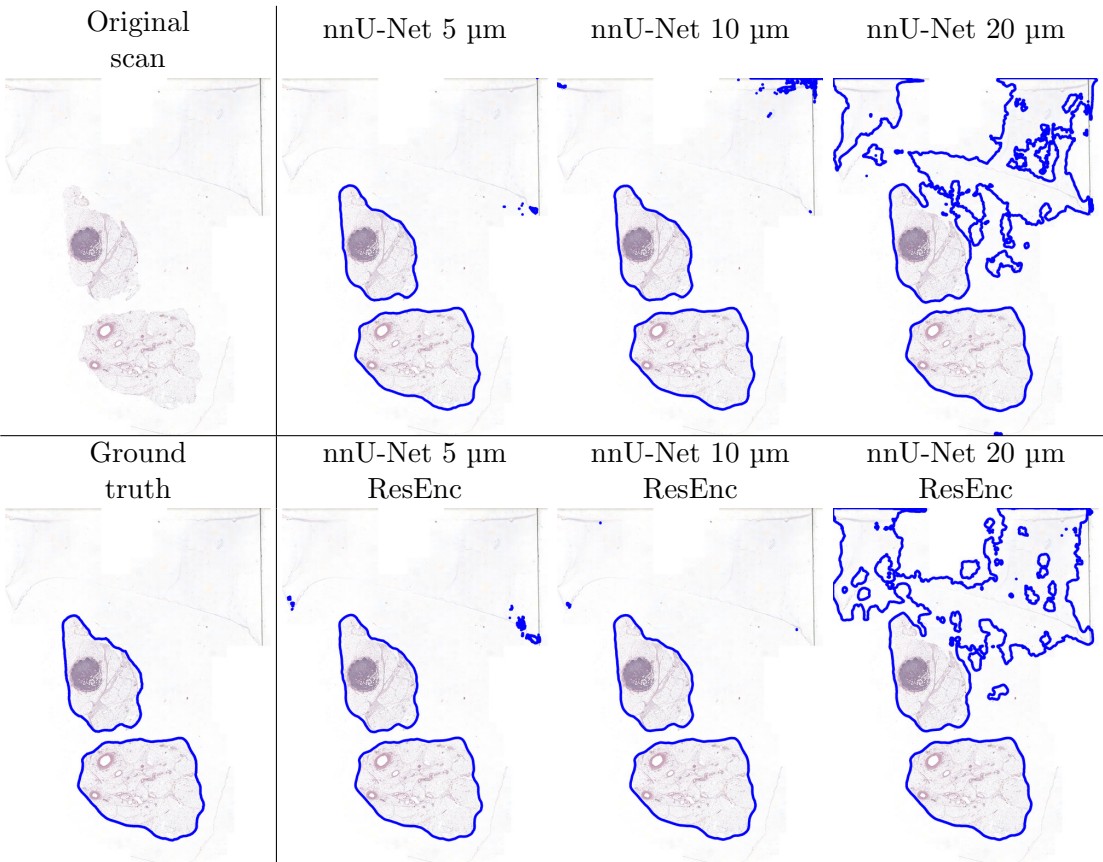

Figure S1: Visual comparison of masks for each resolution for the nnU-Net models. Masks are shown as outlines; the originals are filled. The tissue is taken from a colorectal primary tumour, and a glass edge air bubble is present at the top of the scan.

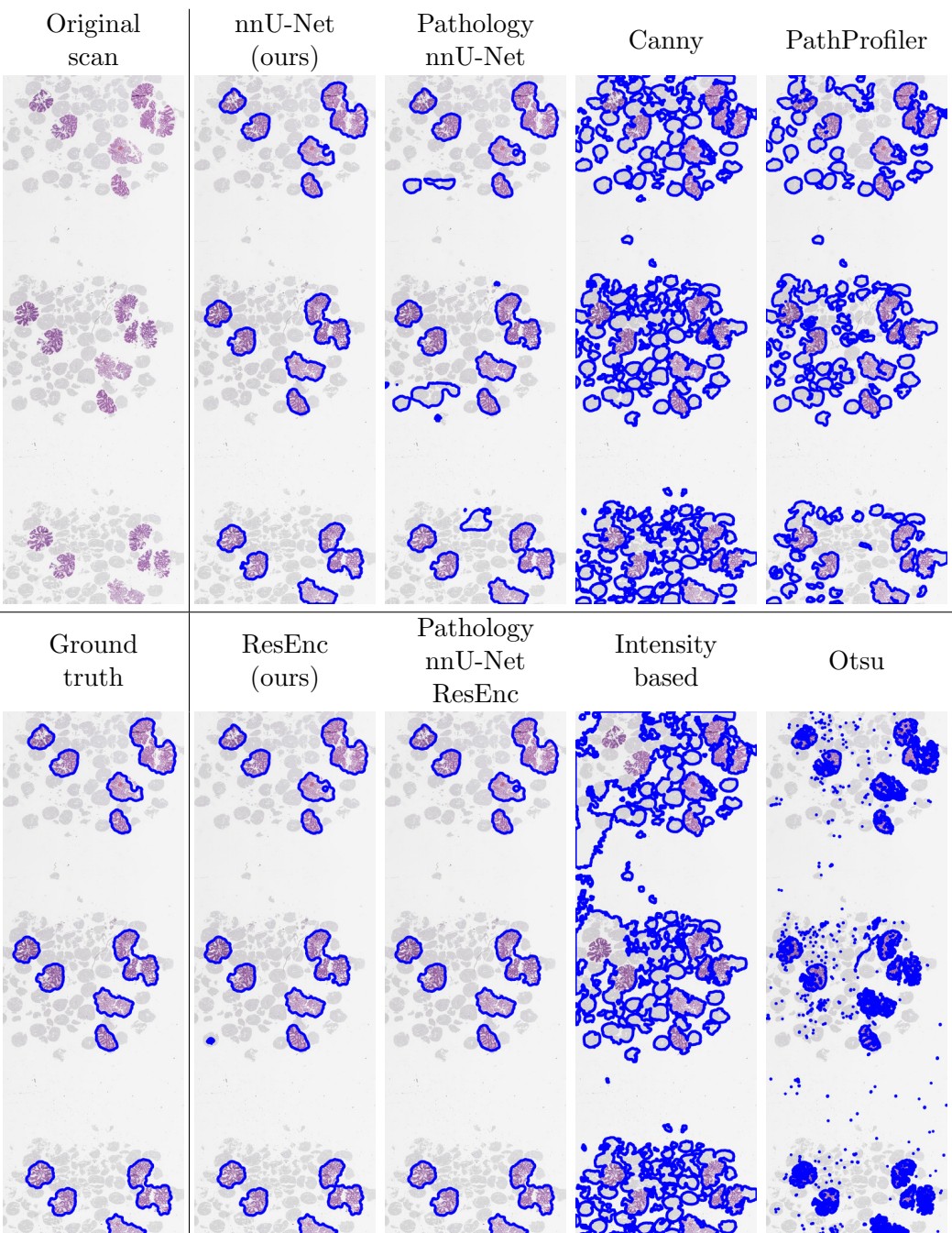

Figure S2: Example result of the different segmentation models presented in this study. Masks are shown as outlines; the originals are filled. The scan show tissue from a low-grade dysplasia colorectal polyp, and the shaded regions are fecal matter.

