# OpenReview forum: "Reliable and Efficient Tissue Segmentation in Whole-Slide Images"
_MICCAI.org/2025/Workshop/COMPAYL — COMPAYL 2025_

### Official Review · Reviewer_BpST · 2025-07-07
**Reliable and Efficient Tissue Segmentation in Whole-Slide Images**

**Rating:** 4
**Confidence:** 4

**Review:**

Summary
This paper introduces a robust tissue segmentation model for Whole-Slide Images (WSIs), leveraging the nnU-Net framework.

Strengths
The proposed model demonstrates comparable segmentation performance and favorable inference times. A significant strength is its superior robustness, exhibiting fewer severe failure cases compared to other deep learning methods evaluated. Furthermore, the authors have made their code publicly available, which is a valuable contribution to the research community.

Weaknesses
While the model shows strong performance, it doesn't statistically outperform other methods. A notable weakness is the lack of clarity regarding the fine-tuning of PathProfiler and Pathology nnU-Net on the authors' specific dataset. Without this crucial detail, readers might be skeptical of the presented comparative results.

---

### Official Review · Reviewer_xNEZ · 2025-07-09
**Simple yet powerful baseline with strong results: nnU-Net for tissue segmentation in WSIs**

**Rating:** 5
**Confidence:** 4

**Review:**

Short summary:
The paper introduces a straightforward yet surprisingly strong baseline for tissue-vs.-background segmentation in whole-slide images (WSIs). Using an unmodified nnU-Net v2 pipeline trained on 28,858 WSIs originating from seven projects and multiple scanner vendors, the authors reach an average Dice of 98+%, outperforming both rule-based baselines and two deep-learning competitors (PathProfiler and a Pathology-adapted nnU-Net). Inference duration demonstrates a favorable speed/accuracy trade-off. Code and pretrained weights are made publicly available.

Strengths:
- Scale & diversity of data – The 28k-slide dataset spans multiple tissue types, scanners, and institutions, with expert-verified masks, boosting ecological validity
- Simplicity & reproducibility – No custom architecture tweaks; the community-standard nnU-Net is used “as-is,” which lowers the barrier for adoption
- Thorough baseline suite – Both classical (Otsu, Canny, intensity-based) and DL (PathProfiler, Pathology nnU-Net) baselines are benchmarked, with clear runtime and accuracy tables
- Practical insight into resolution trade-offs – Experiments at 5, 10, 20 µm explicitly quantify when extra resolution no longer pays off
- Open science – Code, Dockerfile, and model weights are released, facilitating direct reuse in pathology pipelines

Weaknesses:
- Limited external validation – All test slides come from the same seven-project pool; performance on completely unseen public cohorts (e.g., TCGA) is unknown
- Evaluation metrics are single-dimensional – Only Dice is reported; metrics that penalise over/under-segmentation differently (e.g., IoU, Hausdorff, FROC) would paint a fuller picture
- Comparative methods could be broader – Recent lightweight models (e.g., Hover-Net’s tissue branch, SAM-adapters) or stain-agnostic transformers are not included
- Down-sampling assumptions – While 10 µm/px suffices for background masking, some applications (e.g., fat/tumor interface QC) may require finer granularity; the impact of down-sampling on such edge cases is not analysed
- Training cost – 1000 epochs on an RTX 3090 are reported but no wall-clock time or energy usage is provided

Detailed comments & suggestions:
- To improve dataset diversity, could add a leave-one-center-out experiment or test on a public external WSI set to quantify true generalisation
- Since ResEnc adds ~2× runtime for ~0.4 % Dice gain, can explore mixed-precision or smaller encoder widths to see if the accuracy drop can be minimised while matching speed
- Recommended to add nnU-Net commit hash and environment yaml, Zenodo snapshot/DOI and dataset split file

---

### Official Review · Reviewer_7utR · 2025-07-14
**Review of " Reliable and Efficient Tissue Segmentation in Whole-Slide Images"**

**Rating:** 2
**Confidence:** 5

**Review:**

This paper presents a tissue segmentation pipeline using nnUNet-v2 for computational pathology (CPath). The authors train the model on a large dataset of ~30,000 cases and provide the trained weights and code publicly. The project is well-organized, and the manuscript is clearly written and easy to follow. The release of pretrained models is a practical contribution that may benefit the broader CPath community.

Strengths:

The authors make an effort in terms of reproducibility by sharing code and pretrained weights.

The use of a large training dataset (~30,000 slides) strengthens the robustness of the model.




Weaknesses:

Lack of Clarity on Stain Generalizability:
 The manuscript does not specify which staining protocols the model is designed to handle. While the included figure shows a Hematoxylin and Eosin (H&E) stained slide, it remains unclear whether the model generalizes to other common stains such as Immunohistochemistry (IHC). This is a significant omission, as a segmentation model capable of handling IHC or multiple stains would be more impactful.


Limited Novelty:
 The scientific novelty of this work is limited. Tissue segmentation on H&E slides is a well-established problem in CPath, and several existing solutions[1,2] (e.g., SlideSegmenter,) already perform this task effectively. The high Dice scores (>0.98) achieved by baseline models reinforce the notion that this is a relatively solved task. As such, the contribution is primarily infrastructural rather than scientific.

[1]https://researchinformation.umcutrecht.nl/en/publications/tissue-cross-section-and-pen-marking-segmentation-in-whole-slide-
[2]https://pubmed.ncbi.nlm.nih.gov/31871843/